# Thermally activated triplet exciton release for highly efficient tri-mode organic afterglow

Jibiao Jin[1], He Jiang[1], Qingqing Yang[1], Lele Tang[1], Ye Tao[1], Yuanyuan Li[1], Runfeng Chen [1✉], Chao Zheng[1], Quli Fan [1], Kenneth Yin Zhang[1], Qiang Zhao[1] & Wei Huang [1,2✉]

Developing high-efficient afterglow from metal-free organic molecules remains a formidable challenge due to the intrinsically spin-forbidden phosphorescence emission nature of organic afterglow, and only a few examples exhibit afterglow efficiency over 10%. Here, we demonstrate that the organic afterglow can be enhanced dramatically by thermally activated processes to release the excitons on the stabilized triplet state ($T_1^*$) to the lowest triplet state ($T_1$) and to the singlet excited state ($S_1$) for spin-allowed emission. Designed in a twisted donor–acceptor architecture with small singlet-triplet splitting energy and shallow exciton trapping depth, the thermally activated organic afterglow shows an efficiency up to 45%. This afterglow is an extraordinary tri-mode emission at room temperature from the radiative decays of $S_1$, $T_1$, and $T_1^*$. With the highest afterglow efficiency reported so far, the tri-mode afterglow represents an important concept advance in designing high-efficient organic afterglow materials through facilitating thermally activated release of stabilized triplet excitons.

[1] Key Laboratory for Organic Electronics and Information Displays & Jiangsu Key Laboratory for Biosensors, Institute of Advanced Materials (IAM), Jiangsu National Synergistic Innovation Center for Advanced Materials (SICAM), Nanjing University of Posts & Telecommunications, Nanjing 210023, China. [2] Shaanxi Institute of Flexible Electronics (SIFE), Northwestern Polytechnical University (NPU), 127 West Youyi Road, Xi'an 710072, China. ✉email: iamrfchen@njupt.edu.cn; iamdirector@fudan.edu.cn

Organic afterglow materials, which typically show a luminescent lifetime over 0.1 s according to the resolving limit of the naked eyes, have attracted tremendous attention recently[1–3]. By the significant breakthrough of the excited state lifetime tuning of the highly active organic excitons, innovational applications of organic afterglow molecules in various fields such as biological imaging, information storage, sensing, and security protection have been witnessed[4–7]. However, owing to the organic ultralong room temperature phosphorescence (OURTP) nature of organic afterglow through the spin-forbidden radiative decay of the triplet excited states ($T_1$), it is a crucial challenge to achieve high afterglow efficiency[8,9]. Principally, to generate observable afterglow emission, the incorporation of heteroatoms to promote the spin-orbital coupling (SOC) between excited singlet and triplet states for the enhanced intersystem crossing (ISC) and the construction of stabilized triplet excited state ($T_1^*$) to suppress the nonradiative decay are essential (Fig. 1a)[10,11]. Thus, organic afterglow efficiency is doomed to be low, considering that only a small part of photoexcited singlet excitons can be transformed to triplet ones through ISC under weak SOC values of purely organic molecules and the nonradiative transitions dominate the triplet exciton decay with the low phosphorescence efficiency at room temperature[8]. Many efforts have been devoted to address this issue, ranging from promoting the ISC process to efficiently populate $T_1$ and $T_1^*$[11], enhancing the intra/intermolecular interactions to suppress the nonradiative transition[12,13], to incorporating heavy atoms to facilitate both SOC and ISC rates for emission and exciton transformation[13–17]. But, very few attempts succeed in increasing organic afterglow efficiency to 10%, especially in heavy-atom-free molecules[18,19] (Supplementary Fig. 1).

Thermally activated delayed fluorescence (TADF), which can convert triplet excitons to singlet ones via reverse intersystem crossing (RISC) for 100% internal quantum efficiency in harvesting both singlet and triplet excitons for emission, represents the most magnificent advance in recent organic electronics[20,21]. In TADF emitters, by controlling the spatial overlap between the highest occupied molecular orbital (HOMO) and the lowest unoccupied molecular orbital (LUMO), singlet-triplet splitting energy ($\Delta E_{ST}$) can be effectively reduced, rendering efficient RISC to transform the nonemissive triplet excitons to spin-allowed singlet excitons for the delayed fluorescence (Fig. 1b)[21]. Inspired by this policy of TADF in harvesting triplet excitons for emission, we reason that introducing the thermally activated processes into organic afterglow molecules can also significantly improve the luminescent efficiency of $T_1^*$ for high-efficient afterglow emission. Different from that in TADF by transforming $T_1$ excitons to $S_1$ excitons, we need to release the excitons on the stabilized low-lying triplet state of $T_1^*$ to $T_1$ and subsequently transform $T_1$ to $S_1$ for the spin-allowed efficient organic afterglow. Therefore, a shallow trapping depth ($E_{TD}$) should be designed firstly to release the $T_1^*$ excitons by thermal energy fluctuation; secondly, a small $\Delta E_{ST}$ should be established to transform the resulting $T_1$ excitons to $S_1$ excitons through RISC (Fig. 1c). Thus, efficient thermally activated afterglow (TAA) emission can be realized by the cascade thermally activated processes through releasing the long-lived $T_1^*$ excitons and transforming the spin-forbidden phosphorescence of $T_1^*$ and $T_1$ emission to the spin-allowed emission of $S_1$, when both $E_{TD}$ and $\Delta E_{ST}$ were controlled to be sufficiently low.

To prove our design, we prepare such a molecule using difluoroboron β-diketonate and carbazole units in a twisted donor–acceptor–donor (D–A–D) molecular architecture (Fig. 1d).

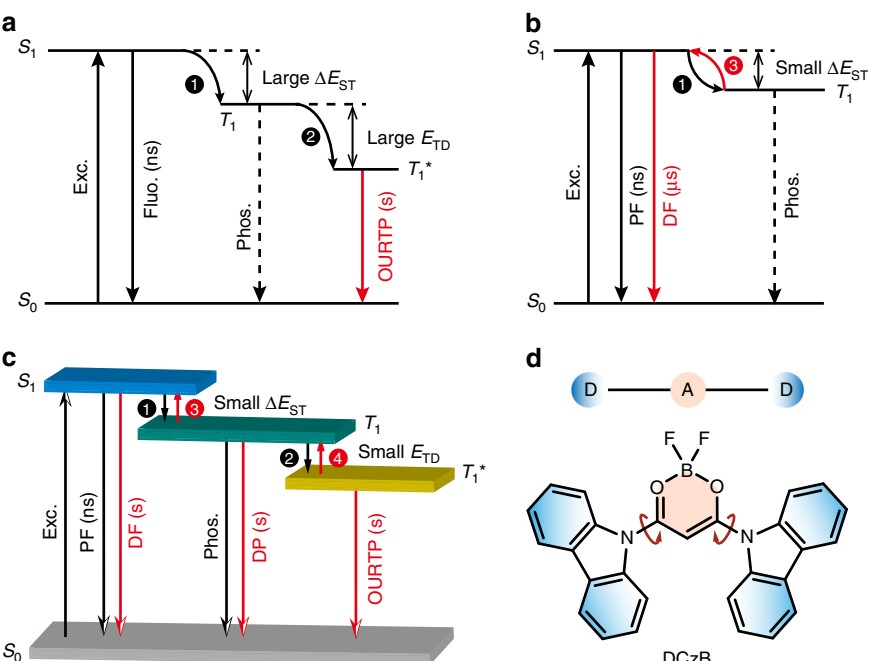

**Fig. 1 A mechanism for improving the organic afterglow efficiency. a** Mechanism of OURTP by constructing $T_1^*$ in organic aggregates. The molecule in the ground state ($S_0$) is excited (Exc.) to the lowest singlet excited state ($S_1$) for fluorescence (Fluo.) with lifetime of several nanoseconds (ns) and ISC (step 1) to populate the lowest triplet excited state ($T_1$) for weak phosphorescence (Phos.); when $T_1$ is stabilized in $T_1^*$ (step 2), the radiative deactivation of $T_1^*$ results in OURTP with lifetime up to seconds (s). **b** TADF mechanism with facile RISC process (step 3) for delayed fluorescence (DF) with lifetime around several microseconds (µs). **c** TAA emission realized by thermally activated exciton release (TAER) (step 4) and RISC (step 3) processes. Firstly, the trapped excitons in $T_1^*$ were released by thermal perturbation to $T_1$ at a small $E_{TD}$ for delayed phosphorescence (DP). Secondly, the released $T_1$ exciton was transformed to $S_1$ exciton for the spin-allowed DF at a small $\Delta E_{ST}$. **d** Design of TAA molecules based on difluoroboron β-diketonate and carbazole in a twisted D–A–D architecture.

Bearing the active nonbonding $p$ electrons, the difluoroboron β-diketonate unit can significantly facilitate the ISC process[22] and more importantly, construct multiple intra/intermolecular hydrogen bond interactions to suppress the nonradiative transitions[23,24]. Carbazole is a widely used group in constructing OURTP molecules with strong tendency to form low-lying $T_1^*$ for the stabilization of triplet excitons in aggregated states[5]. Moreover, difluoroboron β-diketonate is a strong acceptor; the direct bonding to the donor of carbazole will result in strong intramolecular charge transfer (ICT) characteristics for typical TADF features with small $\Delta E_{ST}$ and efficient ISC and RISC processes[25]. Also, the direct connection of carbazole to difluoroboron β-diketonate in a twisted structure will disturb the π–π stacking of carbazole units for perfect H-aggregation, resulting in a small $E_{TD}$ for efficient thermally activated exciton release (TAER) of the stabilized triplet excitons. With this strategy in designing TAA molecules, we realize a tri-mode organic afterglow from the simultaneous emission of $S_1$, $T_1$, and $T_1^*$ with a quantum efficiency up to 45%, which is among the highest efficiencies of organic afterglow molecules reported so far. Our work provides both a new concept and a feasible molecular-design strategy for single-component organic afterglow molecules with high afterglow efficiency, and suggests a promising future of TAA molecules for diverse optoelectronic applications.

## Results

**Synthesis and characterization**. The heavy-atom-free TAA material of difluoroboron 1, 3-di(9H-carbazol-9-yl)propane-1, 3-dione (DCzB) and two control molecules of difluoroboron (Z)-3-(diphenylamino)-3-hydroxy-N,N-diphenyl-acrylamide (DNPhB) and difluoroboron 3-(9H-carbazol-9-yl)-3-oxo-N,N-diphenyl-propanamide (CzBNPh) were facilely prepared through an

efficient two-step reaction (Supplementary Scheme 1) and was fully characterized by [1]H and [13]C NMR spectroscopies, high resolution mass spectrum, and single-crystal X-ray diffraction (Supplementary Figs. 2–9 and Supplementary Table 1).

**Photophysical properties**. DCzB displays a maximum absorption band centered around 380 nm in toluene and 400 nm in film (Fig. 2a) for visible-light excitable photoluminance ascribed to the ICT state of this D–A–D molecule[26–28]. This ICT character becomes more apparent in photoluminescence (PL) spectrum in solution, showing a broad PL band that redshifts with the increase of solvent polarity (Supplementary Fig. 10)[28]. In contrast, the PL in film is blue-shifted with a narrower bandwidth, representing the local-excited (LE) state emission feature. The LE dominated emission leads to the significantly enhanced luminance in solid state for aggregation induced emission (AIE) properties of DCzB (Fig. 2b and Supplementary Fig. 11)[29], where PL enhances gradually with the increase of water fraction ($f_w$). Besides, the PL lifetime also elongates from 4.1 ns to 38.9 ms when $f_w$ increases (Fig. 2b and Supplementary Fig. 12), exhibiting a triplet state-related emission behavior of DCzB that is sensitive to air at room temperature (Fig. 2c and Supplementary Fig. 13). Also, the PL quantum efficiency in solution can be significantly increased from 6.3 to 12.4% by degassing the toluene with argon. From the fluorescence (450 nm) and phosphorescence (475 nm) spectra at 77 K (Supplementary Fig. 14a), the $\Delta E_{ST}$ was inferred to be 0.15 eV, suggesting that DCzB was successfully designed to be a TADF molecule. However, different from the weak phosphorescent emission of typical TADF molecules[30], DCzB shows apparent phosphorescence (Supplementary Figs. 14b, 15). Especially in the aggregated states, both the short-lived (4.0 ns)

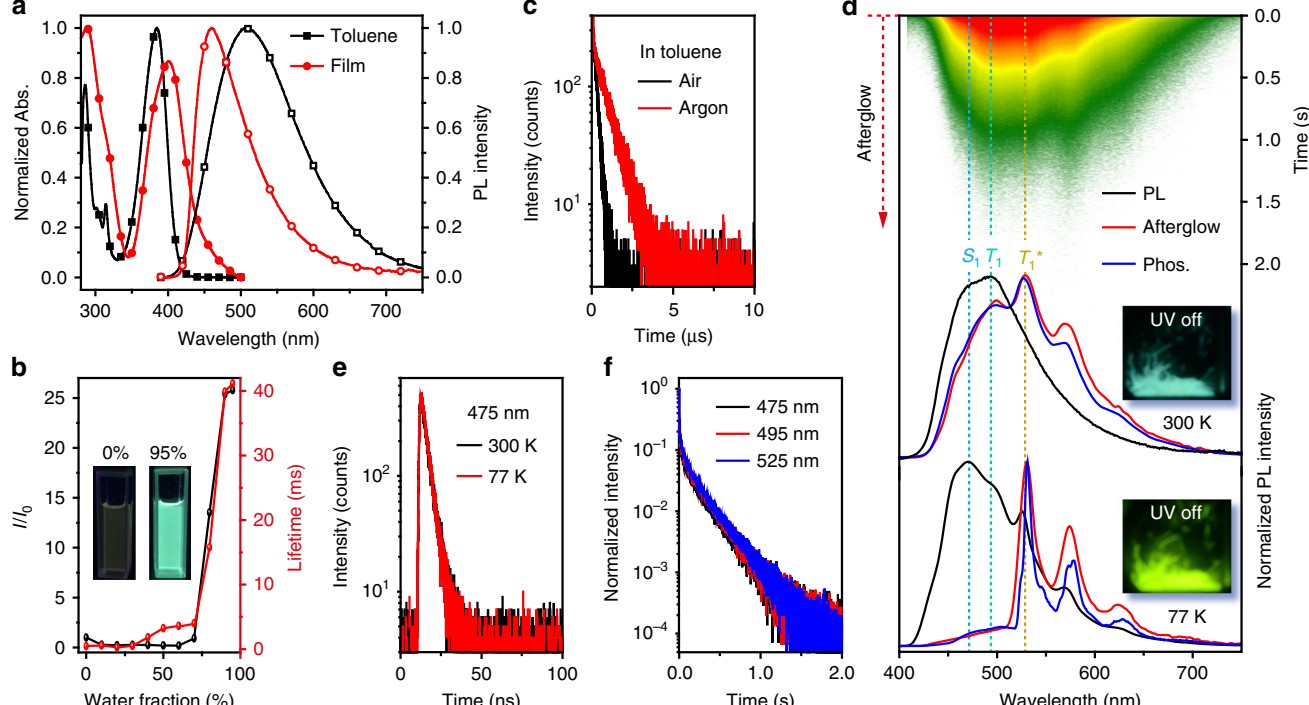

**Fig. 2 Photophysical properties of DCzB. a** Absorption and steady-state PL spectra of DCzB in dilute toluene solution and film states. **b** Plot of PL intensity ratio ($I/I_0$) and PL lifetime versus water fraction in the THF solvent, where $I_0$ is the PL intensity in pure THF and the insets are the photographs of DCzB solution with 0 and 95% water under 365 nm excitation at room temperature. **c** Lifetime decay profiles (510 nm) of DCzB solution in air and argon. **d** Steady-state PL (black line), afterglow (red line, delay 100 ms), and phosphorescent spectra (blue line, delay 5 ms) of DCzB crystal at 300 and 77 K with corresponding photographs taken after the removal of the 365 nm excition source. The upper inset shows the transient PL decay image of the sample recorded at 300 K. **e** Fluorescence decay profiles of DCzB crystal at 300 and 77 K. **f** Afterglow decay profiles of the 475, 495, and 525 nm emission bands of DCzB crystal excited at 380 nm at room temperature. Source data are provided as a "Source Data file".

fluorescence (475 nm) and phosphorescence (495 nm) peaks of DCzB crystal can be recorded in the steady-state PL spectrum at the room temperature (Fig. 2d, e). More interestingly, the fluorescence, phosphorescence, and OURTP emission (525 and 570 nm) of DCzB crystal all exhibit extraordinary ultralong lifetimes over 230 ms at room temperature (Fig. 2f), resulting in simultaneously observed emission bands of $S_1$, $T_1$ and $T_1^*$ in the phosphorescent spectrum (Fig. 2d) for a tri-mode afterglow emission behavior. It should be noted that this tri-mode organic afterglow is insensitive to varied atmospheres (Supplementary Fig. 16) and can be effectively activated by a relatively low power of excitation (<50 μW cm$^{-2}$) within a short period of time (<0.5 s) under ambient conditions (Supplementary Fig. 17). More excitingly, high organic afterglow efficiency up to 45% of DCzB crystal was identified by collecting all the three long-lifetime emission components together at room temperature (Supplementary Table 2 and Supplementary Fig. 18). To the best of our knowledge, this is the highest organic afterglow efficiency of single component purely organic molecules reported so far.

To uncover the reason for the extraordinary involvement of fluorescence for the blue–green organic afterglow, the steady-state PL, afterglow and phosphorescence spectra of DCzB crystal (Fig. 2d) and DCzB (5 wt%)-doped into polymethyl methacrylate (PMMA) film (Supplementary Fig. 19) were also investigated at 77 K. The DCzB-doped PMMA film shows both fluorescence (475 nm, 4.4 ns) and phosphorescence (495 nm, 381 ms), suggesting that the afterglow peaks of 475 and 495 nm in crystal can be ascribed to the radiative decays of $S_1$ and $T_1$ of the isolated single molecules, respectively. The emission peak of 525 nm is undetectable in the doped film, since the $T_1^*$ emission is related to the molecular aggregations[5,31]. In DCzB crystal, owing to the suppressed nonradiative relaxation at low temperatures, all the emission bands are enhanced and the OURTP peaks become observable in the steady-state PL spectrum and the lifetime was elongated to 1.0 s at 77 K (Supplementary Figs. 20, 21). In shape contrast to the afterglow spectrum at room temperature, the fluorescent and phosphorescent bands are absent at 77 K, resulting in a green–yellow color of afterglow emission from only the emission band of $T_1^*$. Indeed, the thermal energy can help the release of the stabilized exciton in $T_1^*$ to $T_1$ and transform the $T_1$ excitons to $S_1$ excitons for the turning of the short-lived fluorescence and phosphorescence to the long-lived afterglow. Therefore, from the corresponding emission peaks, $\Delta E_{ST}$ and $E_{TD}$ of the DCzB crystal can be identified experimentally to be 0.09 and 0.15 eV from the energy differences of $S_1$ and $T_1$ and of $T_1$ and $T_1^*$, respectively.

When the temperature drops from 300 to 80 K, the afterglow emission bands around 475 and 495 nm related to $S_1$ and $T_1$ increase at first, then decrease and finally increase after a critical temperature of 170 K (Fig. 3a, b and Supplementary Fig. 22). The first enhancement is induced by the suppressed nonradiative relaxation under facile TAER and RISC processes, the following lifetime decrease should be due to the much suppressed TAER and RISC processes and the finally enhancement is caused by the fully suppressed nonradiative relaxation similar to TADF at low temperatures. The gradual disappearance of the blue fluorescent and phosphorescent components of the afterglow leads to varied afterglow colors at different temperatures, while the Commission Internationale de L'Eclairage (CIE) of the steady-state PL remains almost constant (Fig. 3c). These experiments provide direct evidences for the effects of TAER in releasing $T_1^*$ excitons and the RISC process to facilitate the delayed fluorescence for TAA.

**Theoretical investigations**. To gain deeper insights into the TAA mechanism in turning fluorescence and phosphorescence to

afterglow, first-principle time-dependent density functional theory (TD-DFT) calculations were performed (Fig. 3d, e). In the isolated single molecular state, HOMO is dominated by the donor moiety of carbazole, whereas the LUMO is mainly located at the acceptor unit of difluoroboron β-diketonate, leading to typically separated HOMO and LUMO distributions of TADF molecules. In the aggregated crystal states, this HOMO and LUMO separation becomes more obvious in the dimer, where they are localized at different molecules. These theoretical findings explain well the reduced $\Delta E_{ST}$ from 0.15 eV in solution to 0.09 eV in crystal as observed experimentally (Fig. 3d and Supplementary Table 3). To investigate the ISC and RISC processes, SOC values between $S_1$ and $T_n$ were also calculated based on single molecular state of DCzB at $T_1$ (Fig. 3e and Supplementary Table 4). Abundant of facile ISC and RISC channels with SOC large than 0.3 cm$^{-1}$ (blue channels)[32] can be found for the facile exciton transformation of DCzB for OURTP. To theoretically understand the trapping and releasing of triplet excitons stabilized by $T_1^*$, the quantum mechanics/molecular mechanics (QM/MM) method was performed to evaluate the electronic properties of the active QM dimer embedded in the aggregated crystal state, while the surrounding molecules were defined as rigid MM part to model the effect of solid-state environment (Fig. 3f). In accordance with the experimental value (0.15 eV), a small theoretical $E_{TD}$ of 0.03 eV was obtained from the triplet state energy difference between the solid state and single molecular state, indicating a relatively weak exciton stabilization effect. This small $E_{TD}$ may be related to the significantly larger torsion angles on the singlet and triplet excited states compared with that in solid state; in dimers, the longer π–π interaction distance on the excited states compared with that on the ground state were observed (Supplementary Scheme 2), suggesting that the photoexcitation of the molecule tends to disturb the π–π interactions in stabilizing triplet excitons by H-aggregation.

**Mechanism insights of TAA**. H-aggregation is another important factor for single-component organic molecules to stabilize their triplet excitons for OURTP[33–35]. Indeed, typical H-aggregates can be identified from the single-crystal structure of DCzB according to the Frenkel exciton theory (Fig. 3g). Also from the single-crystal structure, abundant intra/intermolecular interactions can be revealed, forming a rigid environment to suppress the nonradiative decay for highly efficient emission and to prevent the penetration of oxygen and water from the surrounding environment to quench the emission (Supplementary Fig. 12). Hence, we can propose a possible mechanism for the extraordinary TAA with the state-of-the-art afterglow efficiency (Fig. 3h). Through the facile ISC channels, the photoexcited singlet excitons are readily transformed to $T_1$, which subsequently stabilized by H-aggregation in $T_1^*$. Considering small values of $E_{TD}$ (0.15 eV) and $\Delta E_{ST}$ (0.09 eV), the TAER and RISC processes will be efficient at room temperature to repopulate the $T_1$ and $S_1$ for the delayed phosphorescence and fluorescence. Thus, a tri-mode afterglow emission containing OURTP, phosphorescent afterglow emission (TAA-I), fluorescent afterglow emission (TAA-II) can be realized. More importantly, such thermally activated TAER and RISC processes can turn significant part of spin-forbidden triplet excited state emission to the spin-allowed fluorescence, resulting in dramatically improved organic afterglow efficiency.

To further confirm the TAA mechanism of tri-mode afterglow, we prepared two control molecules of DNPhB and CzBNPh (Supplementary Scheme 1), which have similar molecular structures to that of DCzB but with different photoluminescent properties (Fig. 4). The DNPhB crystal exhibits a bi-mode

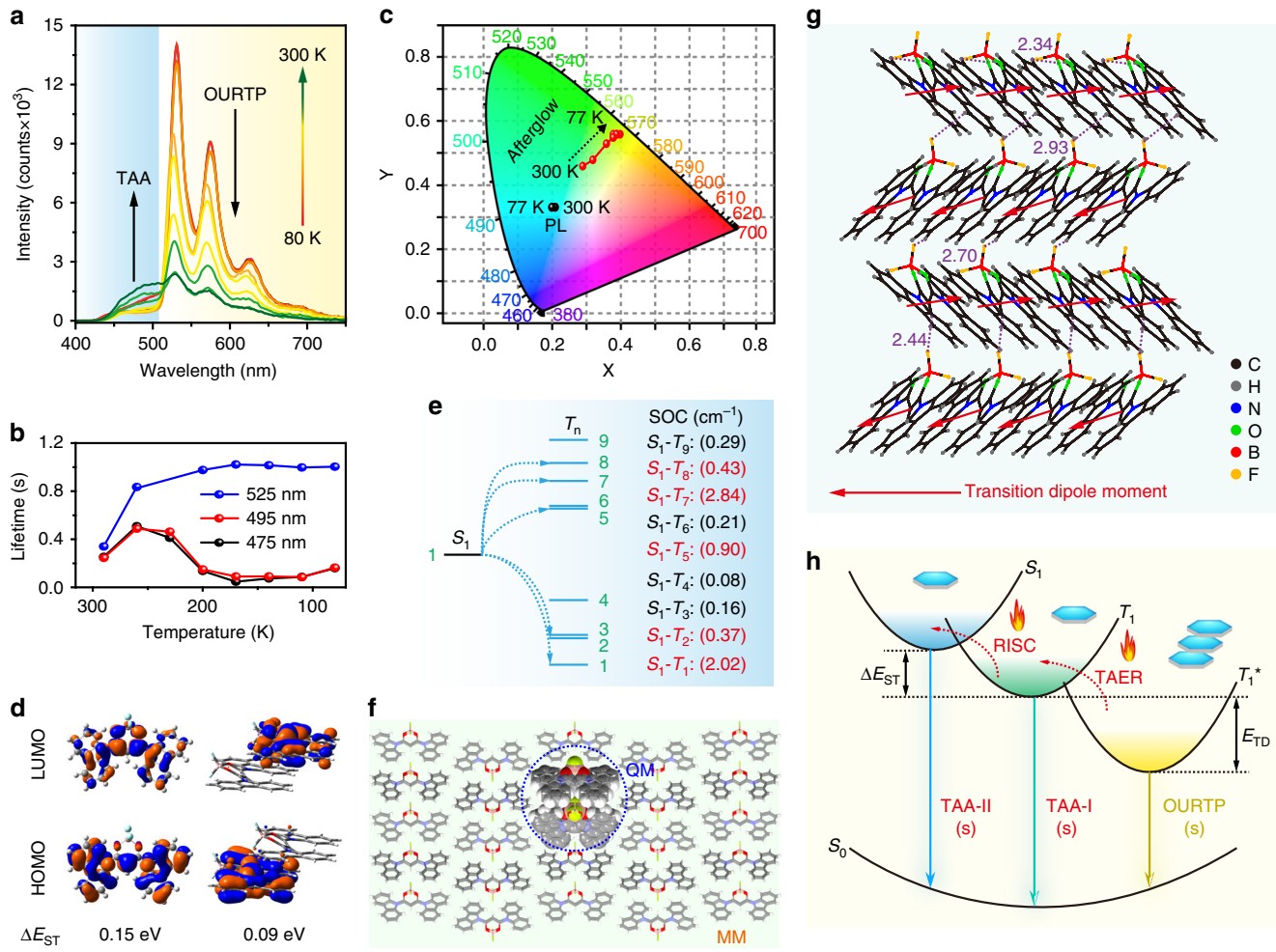

**Fig. 3 Proposed mechanism for the thermally activated organic afterglow. a** Temperature-dependent afterglow spectra from 80 to 300 K of DCzB crystal. **b** Lifetimes of 475, 495, and 525 nm afterglow at different temperatures. **c** CIE 1931 coordinates of steady-state PL and afterglow emission from 80 to 300 K. **d** HOMO and LUMO isosurfaces of DCzB in single molecular and dimer states. **e** TD-DFT calculated energy level diagram and the corresponding SOC constants. **f** QM/MM model of DCzB molecule, the hydrogen atoms were ignored for clarity. **g** Molecular packing and inter/intra-molecular interactions in DCzB crystal. **h** Proposed mechanism of the highly efficient tri-mode afterglow. The excitation wavelength is at 380 nm. Source data are provided as a "Source Data file".

emission of TADF (400 nm, 2.0 ns and 20 ms) and room-temperature phosphorescence (458 nm, 17 ms) from $S_1$ and $T_1$ with $\Delta E_{ST}$ of 0.37 eV but without $T_1^*$ emission for afterglow (Fig. 4a, b). The CzBNPh crystal shows strong afterglow from $T_1^*$ but with very weak afterglow from $S_1$ and $T_1$, since the $\Delta E_{ST}$ (0.22 eV) and $E_{TD}$ (0.33 eV) of CzBNPh are much larger than these of DCzB (Fig. 4c, d). When the temperature drops from 300 to 140 K, the afterglow emission bands of CzBNPh around 430 and 465 nm related to $S_1$ and $T_1$ are gradually reduced and almost disappeared due to the significantly suppressed TAER and RISC processes at low temperatures (Fig. 4c).

## Applications of the TAA molecule.

In light of the significant AIE and visible-light-excitable efficient tri-mode TAA properties, DCzB was tested in afterglow cell imaging and visual temperature detection. For cell imaging, DCzB nanoparticles (NPs) were prepared by a bottom-up approach using an amphiphilic phospholipid of poly(ethyleneglycol)-block-poly(propyleneglycol)-block-poly(ethyleneglycol) (F127) to encapsulate the hydrophobic organic afterglow molecules for good water dispersibility and stability (Fig. 5a). The average particle size was found to be around 100–120 nm (Fig. 5b, c). Owing to the significantly

reduced aggregation size in NPs, slightly blue-shifted absorption spectrum (Fig. 5d) was observed in comparison with that in solid film (Fig. 2a). Nevertheless, there is still an absorption tail around 450 nm, enabling the visible-light-excitable PL properties of the DCzB NPs. Blue-shifted PL and phosphorescent spectra were also observed (Fig. 5d) and the OURTP emission band becomes quite weak in accompany with a decreased lifetime of 49 ms (Fig. 5e). The transparent aqueous solution of DCzB NPs, which shows strong fluorescence due to AIE effects and afterglow emission owing to TAA, was used for the cell imaging. After the DCzB NPs ($0.5 \times 10^{-6}$ M) in phosphate buffer saline was incubated with Hela cell for 2 h, confocal images indicate that the DCzB NPs stain living Hela cancer cells easily, and exhibit strong emission by visible light excitation at 405 nm (Fig. 5f). To take advantage of the long-lifetime emission of DCzB NPs for eliminating the background fluorescence in the living cells entirely, phosphorescence lifetime imaging (PLIM) was carried out (Fig. 5g). Indeed, DCzB NPs in cell display a long-lived emission with averaged lifetime about 500 μs and time-gated luminance images by collecting the photons at long lifetime larger than 100 μs exhibit high signal-to-noise ratios, confirming that DCzB afterglow NPs as a long-lived cell probe is very beneficial for removing the short-lived autofluorescence interference. Further, considering the

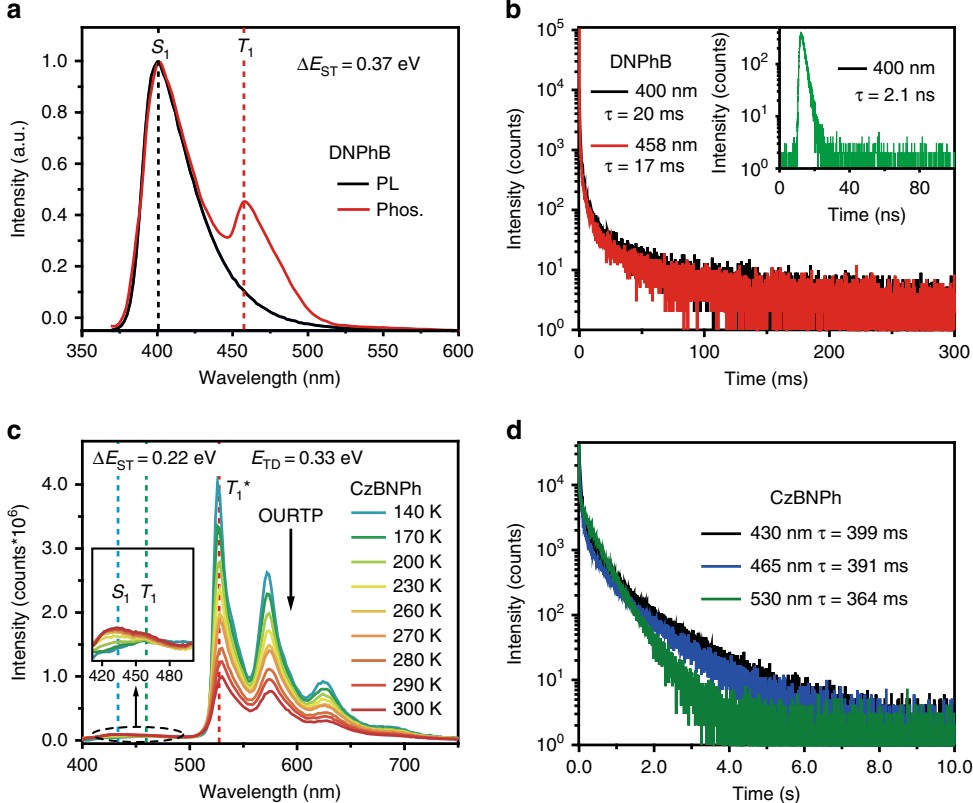

**Fig. 4 Control experiments for confirming the mechanism of TAA. a, b** Steady-state PL and phosphorescent (delay 5 ms) spectra (**a**) and lifetime decay curves (**b**) of DNPhB crystal excited at 330 nm at room temperature. **c, d** Temperature-dependent phosphorescent (delay 10 ms) spectra from 140 to 300 K (**c**) and room temperature lifetime decay curves (**d**) of CzBNPh crystal excited at 390 nm. Source data are provided as a "Source Data file".

strong tri-mode TAA with temperature-dependent afterglow color, we present a potential application of DCzB in multicolor display and visual detection of a specific temperature from 77 to 300 K (Fig. 5h). The pattern "8" is filled with DCzB powder. On the 365 nm excitation, the pattern is blue at different temperatures with negligible color variation, while after the cease of the excitation, the pattern changes from the blue–green afterglow emission at 300 K to green–yellow at 77 K (Fig. 5h), exhibiting a novel multicolor display and quantitative visual detection of temperature (Fig. 5i and Supplementary Fig. 23).

## Discussion

In summary, we have proposed a concept to significantly improve the afterglow efficiency of single-component organic molecules. This concept relies on the simultaneously reduced $E_{TD}$ and $\Delta E_{ST}$ for efficient TAER and RISC processes to release the long-lived $T_1^*$ excitons to $T_1$ by thermal activation and to transform the excitons in spin-forbidden triplet state ($T_1$ and $T_1^*$) to spin-allowed singlet state ($S_1$) for highly efficient afterglow emission. Extraordinarily, a tri-mode afterglow emission with the highest organic afterglow efficiency up to 45% was achieved, in a heavy-atom-free DCzB crystal designed in a twisted D–A–D architecture for the small $E_{TD}$ and $\Delta E_{ST}$. With the efficient and long-lived tri-mode and temperature-sensitive afterglow, high-performance PLIM living cell imaging and multicolour visual sensing of temperature were realized. This study marks a fundamental advance in improving the organic afterglow efficiency for multifunctional applications and offers a general approach for the molecular design of organic room-temperature phosphorescent materials with thermally activated triplet exciton release and transform features.

## Methods

**Preparation of DCzB.** To a 50 mL round-bottom flask charged with 9H-carbazole (1.00 g, 6.0 mmol) was injected 30 mL dry toluene using a syringe under an argon atmosphere. Then, the malonyl dichloride (0.29 mL, 3.0 mmol) was injected to the toluene solution slowly. After stirring at room temperature for 5 min, the reaction was finished and the resulting DCzDCO in white precipitates was collected by filtration under reduced pressure. The precipitates were washed with sufficient acetone to remove the unreacted carbazole to obtain the purified DCzDCO for the next reaction. Yield: 0.90 g of white powder (75%). To a solution of DCzDCO (0.50 g, 1.2 mmol) in 40 mL $CH_2Cl_2$ was added $BF_3 \cdot Et_2O$ (0.46 mL, 3.6 mmol) under a dry argon atmosphere. The mixture was refluxed overnight. Then, the reaction was quenched by cooling to room temperature and washing with water (30 mL). The reaction mixture was extracted with $CH_2Cl_2$ (3 × 30 mL) and the organic layers were collected and dried with anhydrous $Na_2SO_4$. After removing the solvent by vacuum-rotary evaporation, the solid residue was purified by column chromatography (silica gel, 3:1 v/v, PE/$CH_2Cl_2$). Yield: 0.35 g of green powder (65%).

**Measurements.** Ultraviolet/visible (UV/Vis) and PL spectra of DCzB in dilute toluene, $CH_2Cl_2$, tetrahydrofuran (THF), trichloromethane, and solid film were measured on a Lambda 650 S Perkin Elmer UV/VIS spectrophotometer and Edinburgh FLS 980 fluorescence spectrophotometer, respectively. For fluorescence lifetime measurements, a picosecond pulsed light emitting diode (EPLED-380, wavelength: 377 nm; pulse width: 947.7 ps) was used. The phosphorescence spectrum of DCzB in dilute toluene was obtained using an Edinburgh FLS 980 fluorescence spectrophotometer at 77 K in a Dewar vessel with 5 ms delay time after excitation using a microsecond (μs) flash lamp. The microsecond flash lamp produces short, typically a few microsecond, and high irradiance optical pulses for room-temperature phosphorescence measurements in the range from microseconds to seconds. The afterglow spectra, kinetic measurements, lifetime (τ), and time-resolved emission spectra of DCzB crystal were also measured on an Edinburgh FLS 980 fluorescence spectrophotometer. The absolute PL quantum yield was obtained using an integrating sphere.

**Theoretical investigations.** DFT and TD-DFT computations were carried out using Gaussian 09 package. The ground state geometry was optimized by DFT method of B3LYP/6-31 G(d); the optimized stationary point was further characterized by harmonic vibrational frequency analysis to ensure that real local minima were reached. TD-DFT calculations were performed to predict the

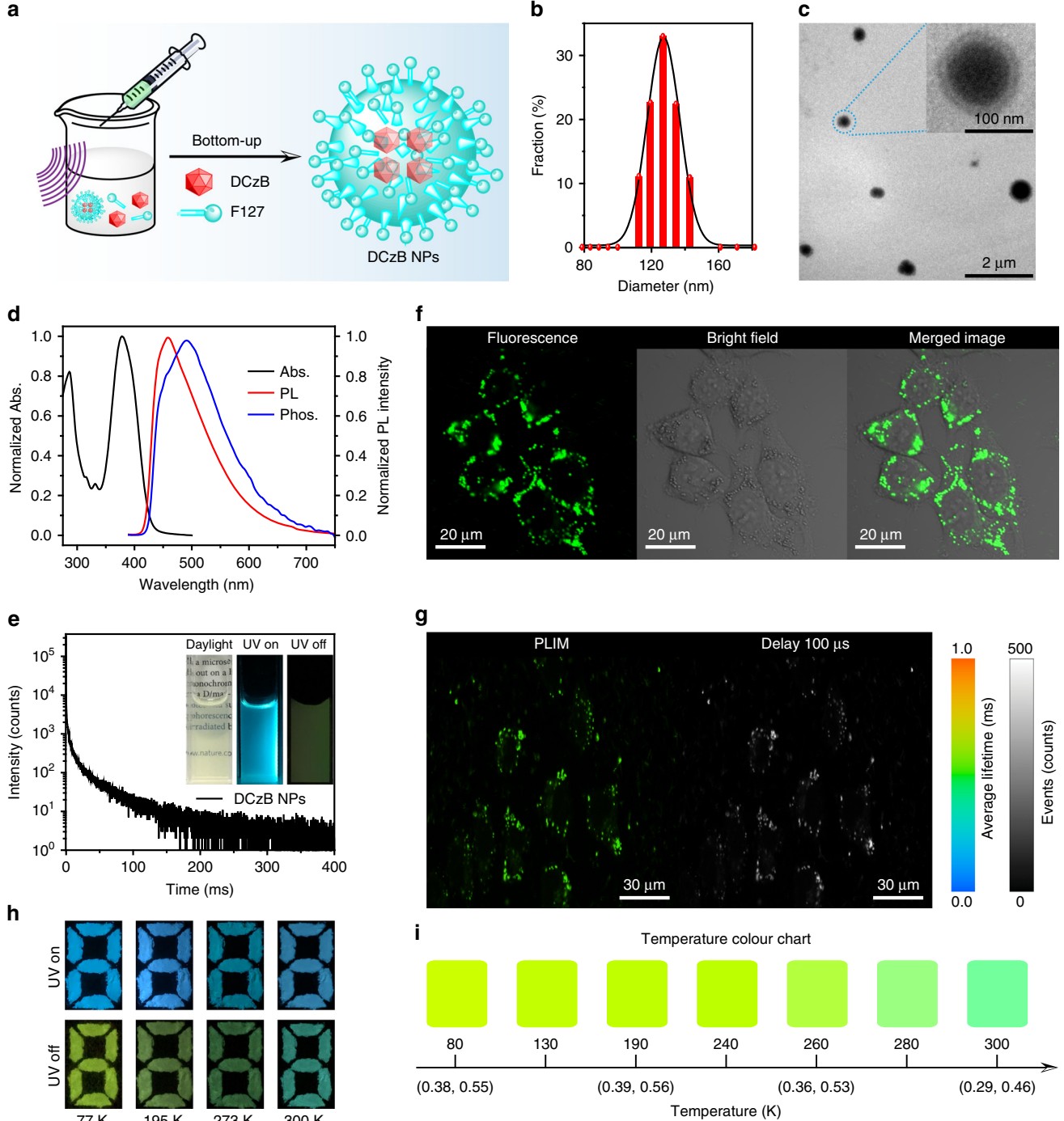

**Fig. 5 Applications in afterglow cell imaging and visual temperature detection. a** Bottom-up preparation of DCzB NPs using F127. **b**, **c** Particle size distribution revealed by dynamic light scattering (**b**) and transmission electron microscope images (**c**). **d**, **e** Absorption (black curve), steady-state PL (red curve), room-temperature phosphorescent spectra (blue curve, delay 5 ms) (**d**) and phosphorescence decay curve (**e**) of the DCzB NPs excited at 380 nm. The inset is the photographs taken under daylight and 365 nm light irradiation (UV on) and after the removal of the excitation (UV off). **f**, **g** Confocal fluorescence images (**f**), PLIM and time-gated images (delayed 100 μs) (**g**) of Hela cells incubated with DCzB nanoparticles at 37°C for 2 h. The collection range is 450–550 nm and the excitation wavelength is 405 nm. **h** Photographs of the pattern before (UV on) and after (UV off) the turning off of the 365 nm UV lamp at 77, 195, 273, and 300 K. **i** Temperature-dependent color chart with corresponding CIE coordinate showing the ability of DCzB crystals in visual sensing of temperatures. Source data are provided as a "Source Data file".

excitation energies in the n-th singlet ($S_n$) and n-th triplet ($T_n$) states on the basis of the optimized ground structure via spin-restricted formalism using B3LYP/6-31 G(d). SOC matrix elements between the singlet excited states and triplet excited states were calculated with quadratic response function methods using the Dalton program[1,36]. The SOC of DCzB was performed at the optimized geometry of the lowest triplet excited state ($T_1$) using B3LYP functional and cc-PVTZ basis set. The QM/MM model was built from the single-crystal structure and was implemented to evaluate the electronic properties of the active QM molecule embedded in the aggregated crystal state, while the surrounding molecules were defined as rigid MM part to model the effect of solid-state environment. The high layer for QM is calculated by the TD-DFT method of B3LYP/6-31 G(d), and the low layer for MM is described by the universal force field enhanced by Coulomb interactions which are in-line with the quantum method.

**Preparation of nanoparticles of DCzB.** F127 (10 mg) and DCzB (1 mg) were dissolved into a THF (1 mL) solution. The mixture was then rapidly injected into Milli-Q water (10 mL) under continuous sonication in a sonicator bath (Branson) for 5 min. Then, THF was evaporated with a gentle nitrogen flow. Finally, the aqueous solution was filtered through a 0.22 μm PVDF syringe driven filter (Millipore). The obtained NP solution was stored in dark at 4 °C.

**Cell imaging.** The cell line Hela (human cervical cancer) was provided by the Institute of Biochemistry and Cell Biology, SIBS, CAS (China) that have been tested for mycoplasma contamination and was authenticated. The cells were grown in DMEM (Dulbecco's modified Eagle's medium) supplemented with 10% fetal bovine serum, 100 mg/mL streptomycin, and 100 U/mL penicillin at 37 °C with 5% $CO_2$. Confocal luminescence imaging was carried out on an Olympus FV1000 laser scanning confocal microscope equipped with a 40 immersion objective len. Under the excitation of 405 nm semiconductor laser, the emission was collected from 450 to 550 nm. PLIM was carried out on the Olympus IX81 laser scanning confocal microscope. The PL signal was detected by the system of the confocal microscope and correlative calculation of the data was performed by professional software which was provided by PicoQuant Company. The light from the pulse diode laser head (PicoQuant, PDL 800-D) with excitation wavelength of 405 nm with a ×40/ NA 0.95 objective lens for single-photon excitation.

**Reporting summary.** Further information on research design is available in the Nature Research Reporting Summary linked to this article.

## Data availability

All relevant data are available upon reasonable request from the corresponding authors. The source data underlying Figs. 2a–f, 3a–c, 3e, 4a–d, 5b and d, e are provided as a Source Data file. CCDC 1941925 [https://www.ccdc.cam.ac.uk/mystructures/structuredetails/6e0754a2-66ab-e911-967e-00505695f620] containing the crystallographic data for DCzB can be obtained free of charge from The Cambridge Crystallographic Data Centre.

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

## Acknowledgements

This study was supported by the National Natural Science Foundation of China (21772095, 91833306, 61875090, and 21674049), 1311 Talents Program of Nanjing University of Posts and Telecommunications (Dingshan), the Six Talent Plan (2016XCL050), and Postgraduate Research & Practice Innovation Program of Jiangsu Province (46030CX18006 and 46030CX17761).

## Author contributions

J.J., R.C. and W.H. conceived the experiments. J.J., H.J., L.T., Y.T. and C.Z. made the synthesis and performed the photophysical property measurements. Q.Y. performed the computational calculations. Y.L., Q.F., K.Y.Z. and Q.Z. performed the bioimaging experiments. J.J., R.C. and W.H. wrote the paper and all authors contributed to the data analysis.

## Competing interests

The authors declare no competing interests.
