## [Peer Review File · Nature Communications]

Reviewers' comments:

Reviewer #1 (Remarks to the Author):

In this work, the authors proposed an interesting strategy based on thermally activated triplet exciton release to significantly enhance the organic afterglow efficiency for tri-mode afterglow. Detailed photophysical investigations and theory calculations give a good support for the thermally activated processes. With their strategy, the highest organic afterglow efficiency has been improved to 45%. Also, this work has expanded the application scope of organic afterglow materials in time-resolved biological imaging and visual temperature sensor. I believe it can attract a broad readership in organic afterglow materials and organic optoelectronic fields. So, I recommend this work to be published on Nature Communications after addressing the following concerns.

Triplet excited state is sensitive to oxygen, the authors proved it by spectra and lifetime in ambient and argon atmosphere, already. However, the PLQY should also be provided both in ambient and argon atmosphere to confirm the triplet state-related emission behavior. The authors disclosed that small ETD might be related to the significantly larger torsion angles on the excited states compared to the ground state, and the photoexcitation of the molecule tended to disturb the π - π interactions in stabilizing triplet excitons by H-aggregation. I suggest the authors to give the optimal configuration of aggregates in excited state instead of single molecular state (Supplementary Scheme 2). It should be much rational. The authors should provide the CIE chromaticity coordinates in Figure 4i to quantitatively identify the visual detection of temperature. The recent references should be cited.

Reviewer #2 (Remarks to the Author):

Realizing organic ultralong room temperature phosphorescence is of crucial importance for the development of highly efficient afterglow materials. In this paper, the authors proposed a new mechanism of tri-mode organic afterglow involving a thermally activated triplet exciton release process to enhance the afterglow efficiency. The photophysical properties of a new TADF molecule, DCzB, are intriguing. However, the proposed concept for the tri-mode emissions from the S_1 , T_1 , and T_1^* states has no definitive proof. The authors should provide more comprehensive sets of experiments supporting this mechanism. The reviewer could not distinguish the essential difference of photophysical processes between the T_1 and T_1^* , and anticipate that the observed phenomenon can be simply explained by the mixing of TADF and phosphorescence (simple bi-mode emissions) from the S_1 and T_1 states. Therefore, I am not sure that the proposed mechanism is actually correct or not.

Based on these comments, I do not think this paper can be strongly recommended for publication in Nature Communication. I expect more comprehensive study will be reported in a more specialized journal in the future.

Reviewer comments:**Reviewer: #1**

In this work, the authors proposed an interesting strategy based on thermally activated triplet exciton release to significantly enhance the organic afterglow efficiency for tri-mode afterglow. Detailed photophysical investigations and theory calculations give a good support for the thermally activated processes. With their strategy, the highest organic afterglow efficiency has been improved to 45%. Also, this work has expanded the application scope of organic afterglow materials in time-resolved biological imaging and visual temperature sensor. I believe it can attract a broad readership in organic afterglow materials and organic optoelectronic fields. So, I recommend this work to be published on Nature Communications after addressing the following concerns.

Response: Thanks for professional comments and kind recommendation of this work. We have revised the manuscript accordingly.

Comments 1. *Triplet excited state is sensitive to oxygen, the authors proved it by spectra and lifetime in ambient and argon atmosphere, already. However, the PLQY should also be provided both in ambient and argon atmosphere to confirm the triplet state-related emission behavior.*

Response: Thanks for the professional suggestion. In the revised manuscript, PLQYs in toluene solution both under ambient and argon atmosphere have been measured. DCzB exhibits a low PLQY in solution, but we can observe that the PLQY is enhanced in argon (12.37%) than in ambient atmosphere (6.31%), which confirms the triplet state-related emission behavior. The related data and discussions have been involved in the revised main text.

Comments 2. *The authors disclosed that small E_{TD} might be related to the significantly larger torsion angles on the excited states compared to the ground state, and the photoexcitation of the molecule tended to disturb the π - π interactions in stabilizing triplet excitons by H-aggregation. I suggest the authors to give the optimal configuration of aggregates in excited state instead of single molecular state (Supplementary Scheme 2). It should be much rational.*

Response: Thanks for the professional comment. In the revised manuscript, we have obtained

the optimal configurations and π - π interaction of aggregates in S_1 and T_1 states by DFT calculations (see supporting information). In dimer, the longer π - π interaction distance on the excited states compared to that on the ground state was observed, which can disturb the π - π interactions significantly. The related data and discussions have been involved in the revised main text. Many thanks.

Scheme S2. The dihedral angles between carbazole (donor) and difluoroboron β -diketonate (acceptor) of **DCzB** in the optimized single molecular geometry at (a) the ground state (S_0), (b) the lowest singlet excited state (S_1) and (c) the lowest triplet excited state (T_1), respectively. The π - π stacking in (d) the single crystal and the optimized dimer structures at (e) S_1 and (f) T_1 states.

Comments 3. *The authors should provide the CIE chromaticity coordinates in Figure 4i to quantitatively identify the visual detection of temperature.*

Response: Thanks for the thoughtful suggestion. The CIE chromaticity coordinates have been provided in Fig. 5j, and were fitted for temperature detection based on the Commission International de l'Eclairage (CIE) coordinate diagram (Figure S23) of DCzB crystal's afterglow colour under different temperatures. Good fitting can be achieved by the fourth order polynomial equation:

$$T = A_1 + A_2 x + A_3 x^2 + A_4 x^3 + A_5 x^4$$

$$T = B_1 + B_2 y + B_3 y^2 + B_4 y^3 + B_5 y^4$$

where T is the temperature from 77 to 300 K, x and y represent the x-coordinate and y-coordinate of afterglow colour respectively, A and B are constants ($A_1 = 0.41722$, $A_2 = -0.00119$, $A_3 = 1.1422 \times 10^{-5}$, $A_4 = -3.07067 \times 10^{-8}$, $A_5 = 3.54869 \times 10^{-12}$, $R^2 = 0.99$; $B_1 = 0.55159$, $B_2 = -2.79942 \times 10^{-5}$, $B_3 = -3.34644 \times 10^{-7}$, $B_4 = 1.56719 \times 10^{-8}$, $B_5 = -5.95813 \times 10^{-11}$, $R^2 = 0.97$), R^2 represents the coefficient of association. Therefore, we can quantitatively identify the visual detection of temperature. Thanks again.

Fig. 5 | Applications in afterglow cell imaging and visual temperature detection. a, Bottom-up preparation of DCzB NPs using F127. **b,c,** Particle size distribution revealed by dynamic light scattering (**b**) and transmission electron microscope images (**c**). **d,e,** Absorption (black curve), steady-state PL (red curve), room-temperature phosphorescent spectra (blue curve, delay 5 ms) (**d**) and phosphorescence decay curve (**e**) of the DCzB NPs. The inset is the

photographs taken under daylight, 365 nm light irradiation (UV on) and after the removal of the excitation (UV off). **f,g** Confocal fluorescence images (**f**), PLIM and time-gated images (delayed 100 μ s) (**g**) of HeLa cells incubated with DCzB nanoparticles at 37 $^{\circ}$ C for 2 h. The collection range is 450-550 nm and the excitation wavelength is 405 nm. **h**, Photographs of the pattern before (UV on) and after (UV off) the turning off of the 365 nm UV lamp at 77, 195, 273, and 300 K. **i**, Temperature-dependent colour chart with corresponding CIE coordinate showing the ability of DCzB crystals in visual sensing of temperature.

Figure S23. The fitting function based on DCzB afterglow emission upon the change of the temperature from 77 to 300 K.

Comments 4. *The recent references should be cited.*

Response: Thanks for the kind reminding. The recent references have been cited in main text (References: *Angew. Chem. Int. Ed.* 2019, **58**, 12102-12106; *Nat. Commun.* 2019, **10**, 5161; *Mater. Today*, 2019, **08**.010.). Thanks again.

Reviewer: #2

Realizing organic ultralong room temperature phosphorescence is of crucial importance for the development of highly efficient afterglow materials. In this paper, the authors proposed a new

mechanism of tri-mode organic afterglow involving a thermally activated triplet exciton release process to enhance the afterglow efficiency. The photophysical properties of a new TADF molecule, DCzB, are intriguing.

Response: Thanks for the positive comments on our work.

However, the proposed concept for the tri-mode emissions from the S_1 , T_1 , and T_1^ states has no definitive proof. The authors should provide more comprehensive sets of experiments supporting this mechanism. The reviewer could not distinguish the essential difference of photophysical processes between the T_1 and T_1^* , and anticipate that the observed phenomenon can be simply explained by the mixing of TADF and phosphorescence (simple bi-mode emissions) from the S_1 and T_1 states. Therefore, I am not sure that the proposed mechanism is actually correct or not.*

Based on these comments, I do not think this paper can be strongly recommended for publication in Nature Communication. I expect more comprehensive study will be reported in a more specialized journal in the future.

Response: We totally understand the concern of this referee. The S_1 and T_1 emissions for fluorescence and phosphorescence have long been documented and widely acknowledged, while the organic afterglow from the T_1^* (stabilized T_1) state was firstly proposed and systematically confirmed in our previous publication (*Nat. Mater.* 2015. **14**. 685-690). In recent years, this proposal has been cited by many groups to explain the organic afterglow mechanism of their materials (*Nat. Photon.* 2019. **13**. 406-411; *Chem. Sci.* 2019. **10**. 7352-7357; *Small.* 2019. e1903270; *Angew. Chem. Int. Ed.* 2019. **58**. 14140-14145; *Angew. Chem. Int. Ed.* 2018. **57**. 8425-8431; *Angew. Chem. Int. Ed.* 2017. **56**. 16302-16307). Indeed, the emission originated from the radiative decay of T_1 and T_1^* all belong to phosphorescence but with different characteristics. The T_1 emission is the phosphorescence of the **isolated single molecule**. The T_1^* emission is the phosphorescence at the **aggregated solid state**. Therefore, the T_1^* emission shows significantly red-shifted peak wavelength and much longer lifetime, resulting in the long-lived organic afterglow emission with lifetime over 100 ms. In a word, we believe the

essential difference of photophysical processes between T_1 and T_1^* is that T_1 emission is a single molecular behavior, while T_1^* emission belongs to a kind of clusterization-triggered emission at solid states (*Adv. Mater.* 2016. **28**. 9920-9940).

In this work, the observed phenomena **cannot** be explained by the mixing of TADF and phosphorescence from the S_1 and T_1 states. The reasons are as follows.

(1) There are three distinguished emission peaks in the afterglow spectra with different luminescent lifetimes, suggesting these three emission bands are related to the three different excited states with different energies and lifetimes. From the steady-state PL, almost the same emission intensity of 475 nm and 495 nm was observed, while 525 nm emission can be hardly identified. However, in the afterglow spectrum (delay 100 ms), 475 nm emission band drops sharply compared to the 495 nm emission owing to the nanosecond lifetime content of the 475 nm emission, and the 525 nm emission becomes the strongest emission band (Supplementary Figure 1). The different intensity decay behaviors of 475, 495, and 525 nm emission at room temperature reflect their different excited-state nature. Therefore, we ascribed this tri-mode afterglow emission at 475, 495, and 525 nm to S_1 , T_1 and T_1^* excited states, respectively.

Supplementary Figure 1. Steady-state PL (black line) and afterglow (red line, delay 100 ms) of DCzB crystal at 300 K.

(2) Temperature-dependent PL spectra and lifetime decay curves show that the emission bands around 475 and 495 nm exhibit downward trend in the whole, on contrary, the afterglow band around 525 nm exhibits upward trend due to the suppressed nonradiative decay, when temperature drops from 300 to 80 K (Fig. 3a,b). Different temperature response behaviors of 475 (S_1), 495 (T_1) and 525 nm (T_1^*) suggest their different temperature-dependent excited-state features. Using the proposed TAA mechanism *via* stabilized triplet exciton releasing, we can perfectly explain this extraordinary emission behavior at various conditions.

Fig. 3 | **a**, Temperature-dependent afterglow spectra of DCzB crystal from 80 to 300 K. **b**, Lifetimes of 475, 495 and 525 nm afterglow at different temperatures.

(3) Indeed, the mixing of TADF and phosphorescence from the S_1 and T_1 states can explain many photophysical phenomena, but it cannot explain this extraordinary tri-mode afterglow. According to this mechanism, there should show only **two** emission bands with quite close emission wavelength corresponding to the radiative decay of the S_1 and T_1 states with small energy difference (small ΔE_{ST}) for TADF. However, in our system, there are **three** emission bands with different wavelengths and their wavelength difference can be up to 50 nm. Moreover, following the mixing of TADF and phosphorescence mechanism, the long lifetime

of the delay component of TADF should **decrease** with reduced temperature by the suppressed RISC process at low temperatures. However, in our tri-mode emission system, when the temperature drops, the 475 and 495 nm emission shows **firstly increased then decreased** lifetimes, while the lifetime of 535 nm emission is always increasing. These phenomena can be hardly explained by the TADF and phosphorescent mechanism, but our TAA mechanism can.

To further support the mechanism of the tri-mode emission related to S_1 , T_1 and T_1^* , we made more sets of experiments. Firstly, we dispersed DCzB (5 wt%) in polymethyl methacrylate (PMMA) to study the single molecular photophysical properties in solid state (Figure S19). The DCzB-doped film shows fluorescence (475 nm, 4.4 ns) and phosphorescence (495 nm, 381 ms) emission bands at 77 K as recorded by the steady-state PL and phosphorescence spectra, which are consistent with the afterglow peaks of 475 and 495 nm in crystal. This measurement was performed at low temperature, because the reverse intersystem crossing can be significantly suppressed and the phosphorescent spectrum (delay 5 ms) can be obtained without the disturbance of fluorescence and TADF emission bands. Therefore, the afterglow peaks of 475 and 495 nm can be ascribed to the radiative decays of S_1 and T_1 of the isolated single molecules, respectively. The emission peak of 525 nm is undetectable in the doped film, confirming that the T_1^* emission is closely related to the aggregated molecular states after clusterization. The nature of excited states for 475 (S_1), 495 (T_1) and 525 nm (T_1^*) emission are essentially different from each other and we have summarized these differences in Supplementary Table 1.

Figure S19. (a) Steady-state photoluminescence (black and red curves) and phosphorescence (green and blue curves, delay 5 ms) spectra of DCzB crystal at 300 K and DCzB (5 wt%)-doped film in PMMA at 77 K. Lifetime decay curves at (b) 475 and (c) 495 nm of DCzB (5 wt%)-doped film in PMMA at 77 K.

Supplementary Table 1. Photophysical property differences of the tri-mode afterglow emission bands.

Afterglow emission bands of DCzB crystal			
Emission peak	475 nm (S_1)	495 nm (T_1)	525 nm (T_1^*)
Lifetime	4.0 ns; 204 ms	198 ms	196 ms
Temperature response	First increase then decrease at elevated temperatures		Decrease at elevated temperatures

Secondly, we prepared two control molecules of DNPhB and CzBNPh, which have similar molecular structures to that of DCzB but with different photoluminescent properties (Fig. 4). The related synthesis and structure characterizations have been described in supporting information (Scheme S1 and Figure S5-S8). The DNPhB crystal exhibits a bi-mode emission of TADF (400 nm, 2.1 ns and 20 ms) and room-temperature phosphorescence (458 nm, 17 ms) from S_1 and T_1 with ΔE_{ST} of 0.37 eV but without T_1^* emission for afterglow (Fig. 4a,b). These results suggest again that the mixing of TADF and phosphorescence (bi-mode emission) cannot explain the afterglow emission behavior of DCzB and would result in only room-temperature phosphorescence without long-lived (lifetime > 100 ms) afterglow emission.

The CzBNPh crystal shows strong afterglow from T_1^* but with very weak afterglow from S_1 and T_1 , since the ΔE_{ST} (0.22 eV) and E_{TD} (0.33 eV) of CzBNPh are much larger than these of DCzB. When the temperature drops from 300 K to 140 K, the emission intensities of CzBNPh around 430 and 465 nm related to S_1 and T_1 are gradually reduced and almost disappeared in phosphorescent spectra due to the significantly suppressed TAER and RISC processes at low temperatures (Fig. 4c). Therefore, these control molecules further clarify the effects of ΔE_{ST} and E_{TD} in manipulating the organic afterglow, providing solid evidence to support the thermally activated afterglow (TAA) mechanism of the tri-mode afterglow emission.

Scheme S1. The synthetic route to prepare DCzB, CzBNPh and DNPhB. The reaction conditions:

(i) toluene, room temperature; (ii) $\text{BF}_3 \cdot \text{Et}_2\text{O}$, CH_2Cl_2 , room temperature.

Fig. 4 | Control experiments for confirming the TAA mechanism. a,b Steady-state PL and phosphorescent (delay 5 ms) spectra (a) and lifetime decay curves (b) of DNPb crystal excited at 330 nm at room temperature. c,d Temperature-dependent phosphorescent (delay 10 ms) spectra from 140 to 300 K (c) and room-temperature lifetime decay curves (d) of CzBNPh crystal excited at 390 nm.

All in all, we hope we have provided comprehensive sets of experiments to support the S_1 , T_1 , and T_1^* tri-mode afterglow emission mechanism for the thermally activated afterglow (TAA). With the novel TAA concept through stabilized triplet exciton releasing, thus developed organic afterglow molecule in a torsional D-A architecture exhibits the state-of-the-art afterglow efficiency up to 45%. This efficiency exhibits large improvements of 30% to 215% compared to the recent high afterglow efficiency of 34.5% [Chem. 1. 592-602 (2016)], 31.2% [Nat. Photon. 13. 406-411 (2019)] and 14.3% [Nat. Commun. 10. 1595 (2019)] reported in single-component heavy-atom-free materials. To the best of our knowledge, this is the highest afterglow efficiency of purely organic molecules reported so far and is the first observation of tri-mode organic afterglow. We believe that it will set off a wave in the organic afterglow for highly efficient organic optoelectronic materials, promoting significantly the

technological advance in expanding the scope of luminescent materials for optoelectronic applications.

A list of changes made:

- (1) The author list has been updated according to their contribution in the revised manuscript.
- (2) PLQYs in toluene solution both under ambient and argon atmosphere have been measured and discussed in the revised manuscript.
- (3) Optimal configurations of aggregates in excited states have been investigated and discussed.
- (4) The CIE chromaticity coordinates have been provided in Figure 5i and fitting curve has been provided in supporting information to quantitatively identify the visual detection of temperature.
- (5) The recent references have been cited.
- (6) The photophysical properties of DCzB (5 wt%) doped into PMMA have been measured and discussed in the revised manuscript.
- (7) Two control molecules of DNPhB and CzBNPh have been prepared. The related synthesis and structure characterizations have been described in supporting information. The photophysical properties have been measured and illustrated in a new figure of Figure 4 to investigate the photophysical mechanism of the tri-mode organic afterglow emission in the revised manuscript.

REVIEWERS' COMMENTS:

Reviewer #1 (Remarks to the Author):

The authors have fully addressed the comments proposed in the first review. In this case, I recommend acceptance of it.

Reviewer #2 (Remarks to the Author):

The revision has been made adequately. So the paper can be recommended for publication.

Reviewer comments:

Reviewer: #1

The authors have fully addressed the comments proposed in the first review. In this case, I recommend acceptance of it.

Response: Thanks for kind recommendation of this work.

Reviewer: #2

The revision has been made adequately. So the paper can be recommended for publication.

Response: Thanks for kind recommendation of this work.